# Changes in horizontal strabismus after inferior rectus muscle recession with or without nasal transposition in thyroid eye disease: A retrospective, observational study

**Aric Vaidya, Hirohiko Kakizaki, Yasuhiro Takahashi** *

Department of Oculoplastic, Orbital & Lacrimal Surgery, Aichi Medical University Hospital, Aichi, Japan

* yasuhiro_tak@yahoo.co.jp

## Abstract

Since the inferior rectus muscle (IRM) is a secondary adductor, it is expected to commonly observe esotropia in thyroid-associated inferior rectus myopathy, but this can be improved after the IRM recession. However, variable outcomes regarding the changes in horizontal strabismus after IRM recession ± IRM nasal transposition have been encountered in patients with thyroid eye disease (TED). We, therefore, examined the changes in 62 patients with TED in this retrospective, observational, comparative study. The patients were classified into 3 groups based on the results of postoperative changes in horizontal strabismus: Groups A (reduced esotropia), B (unchanged esotropia), and C (increased esotropia). Consequently, Groups A, B, and C included 23 (38.7%), 11 (17.7%), and 27 (43.5%) patients, respectively. In the multivariate linear regression analysis, the angle of preoperative esotropia (P < 0.001) and the amount of IRM nasal transposition (P = 0.049) were significant predictors of postoperative changes in horizontal strabismus. The results of our study will be helpful to ophthalmologists for formulating an effective preoperative surgical plan.

**Data Availability Statement:** All relevant data are within the paper and its Supporting Information file.

## Introduction

Inferior rectus muscle (IRM) is frequently involved in thyroid eye disease (TED) [1–5]. IRM recession is a commonly performed surgical procedure in the correction of restrictive hypotropia in patients with TED, as the muscle recession releases a fibrous contracture of the IRM [4–7]. As excyclotropia is frequently accompanied by thyroid-associated inferior rectus myopathy [1,8–11], additional nasal transposition of the fibrous IRM is a useful treatment in such cases [12,13].

The IRM is a secondary adductor because it runs obliquely at an angle of 23 degrees against the sagittal plane [14]. When it contracts due to cicatricial changes, the eye gets drawn medially. It is, therefore, expected to commonly encounter esotropia in patients with thyroid-associated inferior rectus myopathy, but this can be improved after IRM recession.

However, we have experienced variable outcomes regarding the changes in horizontal strabismus after IRM recession ± IRM nasal transposition in patients with TED. Moreover, in

**Funding:** The authors received no specific funding for this work.

**Competing interests:** The authors have declared that no competing interests exist.

some cases, contrary to the expected postoperative reduction in esotropia, we happened to experience increased esotropia following surgery. While formulating a preoperative surgical plan, the understanding of the tendency of postoperative changes in esotropia will be helpful to ophthalmologists for making a judgement regarding the necessity of concomitant horizontal rectus muscle surgery. Previous studies had shown the changes in horizontal strabismus and AV-pattern strabismus after the recession of the superior rectus muscle (SRM), which is another type of secondary adductor [15,16]. However, to our knowledge, there has not been any report on the changes in horizontal strabismus after IRM recession ± IRM nasal transposition in patients with TED. As the outcomes of strabismus surgery in TED patients are less reproducible than those in patients without TED, it is of utmost importance for the clinicians to understand the factors that influence strabismus in these patients [5,11,17–19].

Hence, in this study, we investigated the changes in horizontal strabismus after IRM recession ± IRM nasal transposition and their predictive factors in patients with TED.

## Materials and methods

### Study design, patients, and diagnosis of TED

This retrospective, observational, comparative study included patients with TED who underwent unilateral IRM recession with or without IRM nasal transposition for diplopia in the primary position at Aichi Medical University Hospital by two of the authors (HK and YT) or registered doctors supervised by one of the authors (YT), between April 2012 and June 2019. The exclusion criteria included a history of strabismus surgery, missing clinical data, follow-up time of < 3 months, and concomitant neuro-ophthalmologic disorders. Patients who underwent orbital decompression before IRM recession, temporal IRM transposition, bilateral IRM recession, and concomitant strabismus surgery on other extraocular muscles were also excluded from the analysis.

A diagnosis of TED was based on the presence of at least one of the characteristic eyelid signs (eyelid fullness, eyelid retraction, and/or eyelid lag) and the presence of thyroid autoimmunity [20]. All the patients showed restrictive hypotropia caused by enlargement of the IRM without muscle tendon involvement, which also supported the diagnosis of TED [21]. They were in controlled state as euthyroid at the time of surgery, and in the static or chronic "burnout" phase of TED. We judged the condition of each patient's disease using the clinical activity score (CAS), as well as by the presence or absence of inflammation in the extraocular muscles, as revealed by the magnetic resonance imaging (MRI). The CAS was calculated using 7 parameters: retrobulbar discomfort, pain on eye movement, eyelid erythema, eyelid swelling, conjunctival injection, chemosis, and swollen caruncle [22]. When the patients had CAS of less than 3 points, their TED was defined as being in the static or chronic "burnout" phase [22].

### Ethics approval

This study was approved by the Institutional Review Board (IRB) of Aichi Medical University Hospital (approval No. 2017-H234) and adhered to the tenets of the Declaration of Helsinki and its amendments. The IRB granted a waiver of informed consent for this study on the basis of the ethical guidelines for medical and health research involving human subjects established by the Japanese Ministry of Education, Culture, Sports, Science, and Technology and the Ministry of Health, Labour, and Welfare. The waiver was granted because the study was a retrospective chart review, not an interventional study, and because it was difficult to obtain consent from patients who had been treated several years prior to the study. Nevertheless, at the request of the IRB we published an outline of the study, which is available for public viewing on the Aichi Medical University Hospital website. This public posting also gave patients

the opportunity to decline participation, although none of the patients did so. Personal identifiers were removed from all records prior to data analysis. Written informed consent for the publication of S1 Fig has been obtained from the patient.

## Data collection

The following data were collected: patient age, sex, CAS, surgical side, MRI findings, smoking status, history of corticoid therapy and/or orbital radiotherapy, amounts of IRM recession and nasal transposition, and preoperative and postoperative angles of ocular deviation.

We asked all the patients the number of cigarettes smoked per day. All smokers were current smokers at the time of first examination, while all non-smokers had never smoked. The smoking status was classified by the number of cigarettes smoked per day, according to a report by Pfeilschifter and Ziegler as follows: 0, no smoking; 1, <10 cigarettes/day; 2, 10–20 cigarettes/day; and 3, >20 cigarettes/day [23].

The angle of ocular deviation was measured using a synoptophore (Clement Clarke International Ltd., Edinburgh, UK) 1 day before surgery and 3 months after surgery by a single trained orthoptist in the same manner as in our previous studies [12,19]. The patient's head was positioned upright, and the instrument set such that the fixating eye was the eye on which surgery was planned. One of the two arms of the synoptophore was fixed at 0˚, 15˚ upward gaze, or 15˚ downward gaze. We used two slides: a black circle with a cross-shaped blank and a black cross (L-25G; Inami, Tokyo, Japan). The black circle slide was fixed in the arm of the fixating eye, and the patient was asked to move the black cross until it was positioned appropriately within the circle. We then recorded the angle at which this was achieved. The deviation angles were measured in the primary position, 15˚ upward gaze, and 15˚ downward gaze.

## MRI

MRI was performed using a 1.5-Tesla scanner (Magnetom Abant™; Siemens Healthcare, Erlangen, Germany), with the patients in the supine position with their eyes fixed in the primary position. The cross-sectional areas of the extraocular muscles were measured on a T1-weighted MRI image (repetition time: 500 ms, echo time: 10 ms, field of view: 140 × 140 mm, matrix: 256 × 220, section thickness: 3 mm with a 0.6 mm gap between slices) by one of the authors (YT), using the freehand measuring tool available in the MRI viewer (ShadeQuest/ViewR™; Yokogawa Medical Solutions Corporation, Tokyo, Japan) (Fig 1). The cross-sectional

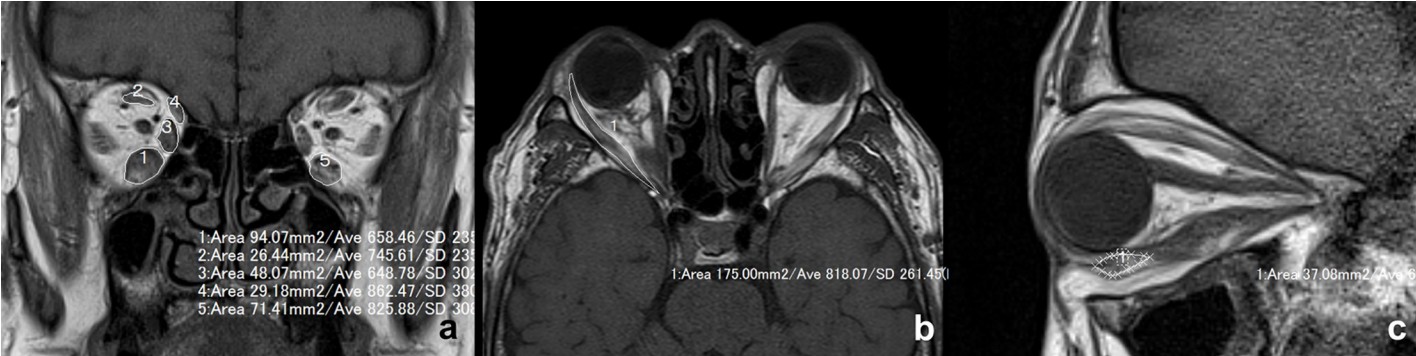

**Fig 1. Measurements of the cross-sectional areas of the extraocular muscles. a.** The cross-sectional areas of the ipsilateral inferior rectus, medial rectus, superior rectus, and superior oblique muscles, and the contralateral inferior rectus muscle at the largest points are measured on a coronal plane. **b.** The cross-sectional area of the ipsilateral lateral rectus muscle is measured on an axial image showing the whole length of the muscle. **c.** The cross-sectional area of the ipsilateral inferior oblique muscle is measured on the sagittal image through the optic nerve.

areas of the ipsilateral IRM, SRM, medial rectus muscle (MRM), and superior oblique muscle (SOM), and contralateral IRM at the largest points were measured on coronal planes (Fig 1A), and that of the ipsilateral lateral rectus muscle (LRM) was measured on an axial image showing the whole length of the LRM (Fig 1B). The cross-sectional area of the ipsilateral inferior oblique muscle (IOM) was also measured on the sagittal plane through the optic nerve, based on our previous study (Fig 1C) [24].

## Surgical procedure

Surgery was performed under either general or local anesthesia. A perilimbal conjunctival incision, with radial relaxing incisions, was made in the inferior quadrant. A muscle hook was used to secure the IRM at its insertion, and the Tenon's capsule around the IRM was thoroughly dissected using cotton swabs. As the tendon width varies between individuals, the width of the IRM tendon was measured at the scleral insertion using a caliper. The IRM tendon was secured using locking 8–0 polyglactin sutures at two points (Vicryl®; Johnson and Johnson Company, New Brunswick, NJ, USA) 1 mm posterior to the globe insertion because the tip thickness of the muscle hook was 1 mm (S1 Fig). Next, the IRM was detached from its insertion. In patients with a simple IRM recession, the sutures were fixed to the sclera 1 mm posterior to the point that was estimated based on the preoperative angle of hypotropia (S1 Fig). The recession of the IRM was calculated as follows: 2.3˚ of hypotropic angle per 1 mm IRM recession [19]. In patients with both IRM recession and nasal IRM transposition, we transposed the IRM nasally along the spiral of Tillaux; we did so concomitantly with IRM recession. The amount of nasal IRM transposition was preoperatively calculated based on the preoperative angle of excyclotropia and the measurement result of the tendon width as follows: 8˚ of excyclotropic angle per one IRM tendon width transposition + 0.4˚ of excyclotropic angle per 1mm IRM recession [12]. The IRM tendon was fixed to the sclera using 8–0 polyglactin sutures at four to six points in total to prevent the slippage of the muscle. Finally, the conjunctiva was closed using 8–0 polyglactin sutures.

## Statistical analyses

Patient data and measurement results were expressed as means ± standard deviations. Cases were classified into the following 3 groups, based on the postoperative changes in the magnitude of horizontal strabismus: Group A (patients in whom esotropia was reduced), Group B (patients in whom esotropia was unchanged), and Group C (patients in whom esotropia was increased). To evaluate the pattern strabismus, the horizontal deviation angle measured in a 15˚ downward gaze was subtracted from that measured in a 15˚ upward gaze. The values were expressed as either positive or negative for evaluation of the magnitude of A- and V-pattern strabismus, respectively. As we roughly set the measurement of 2 prism dioptres corresponded to 1˚, subtraction values of 5˚ and -7.5˚ or greater was considered as clinically significant A- and V-pattern strabismus, respectively [25]. Comparisons between the groups were conducted via one-way ANOVA following Tukey-Kramer post-hoc test or chi-square test. The changes in the magnitude of strabismus were compared between pre- and postoperative measurement results using paired $t$-test in total and using Wilcoxon signed-rank test in each group. Univariate and following multivariate linear regression analyses with stepwise variable selection were performed to identify the factors influencing changes in horizontal strabismus. When esotropia was reduced after surgery, postoperative changes in the magnitude of horizontal strabismus were expressed as positive values. On the contrary, when esotropia was increased after surgery, postoperative changes in the magnitude of horizontal strabismus were expressed as negative. The predictive variables investigated were patient age, smoking status, history of

corticoid therapy and/or orbital radiotherapy, amounts of IRM recession and nasal transposition, angle of preoperative esotropia, and the cross-sectional areas of the extraocular muscles. History of previous corticoid therapy/orbital radiotherapy was expressed using a binary system (a dummy variable; 0 = with no history, 1 = with a history). The angles of preoperative hypotropia and excyclotropia were not included in this multivariate linear regression analysis because these factors can be statistical confounders that might influence both the dependent variable and the amounts of IRM recession and IRM nasal transposition, respectively. The relationship between the amounts of IRM recession and IRM nasal transposition was analyzed using Pearson's correlation coefficient because the effect of IRM recession can offset that of IRM nasal transposition regarding postoperative changes in the magnitude of esotropia. All statistical analyses were performed using SPSS™ version 22 software (IBM Japan, Tokyo, Japan). Two-tailed $P$ values $< 0.05$ were deemed to indicate statistical significance.

## Results

Patient data and measurement values are shown in Table 1. Sixty-two patients (24 men and 38 women; 30 right and 32 left; age at surgery, 60.7 ± 9.5 years) with TED underwent IRM recession, with or without nasal IRM transposition. Forty-eight patients underwent both IRM recession and nasal IRM transposition; in contrast, the other 14 patients underwent only IRM recession. Forty-six patients were non-smokers and 27 patients had undergone corticoid therapy and/or orbital radiotherapy before the IRM recession. All the patients met the inclusion criteria and none of the patients were excluded from this study.

Esotropia was reduced (Group A), unchanged (Group B), and increased (Group C) after IRM recession ± IRM nasal transposition in 24, 11, and 27 patients, respectively. The male-to-female ratio was significantly different among the groups (P = 0.039). The ratio of patients who had undergone corticoid therapy and/or orbital radiotherapy before the IRM recession tended to be larger in Group B (P = 0.082). The angle of preoperative esotropia was significantly larger in Group A than in Group C (P = 0.015). The cross-sectional area of the ipsilateral SOM was thicker in Group C than in Group A (P = 0.047).

The magnitude of pattern strabismus was not different between the pre- and postoperative ones in total (P = 0.585) and Groups A (P = 0.737), B (P = 0.796), and C (P = 0.687), respectively. As for the intergroup difference, patients in Group C showed V-pattern postoperatively, compared to those in Group A (P = 0.027). Only 1 patient in Group A showed clinically significant A-pattern strabismus (5°), which was corrected after surgery. On the other hand, only 1 patient in Group C who had not shown pattern strabismus before surgery demonstrated V-pattern strabismus (-8°) after surgery.

The results of the univariate and multivariate linear regression analyses are shown in Table 2. The univariate analysis showed that the postoperative changes in esotropia were correlated with the angle of preoperative esotropia (P < 0.001), the amount of IRM nasal transposition (P = 0.041), and the cross-sectional area of the SOM (P = 0.012). In the multivariate linear regression analysis, the angle of preoperative esotropia (P < 0.001) and the amount of IRM nasal transposition (P = 0.049) were significant predictors of the postoperative changes in the magnitude of horizontal strabismus. Patient age, smoking status, history of corticoid therapy and/or orbital radiotherapy, the amount of IRM recession, and the cross-sectional areas of the extraocular muscles were deleted from the regression model by stepwise procedure (P > 0.050). The predictive equation for postoperative changes in horizontal strabismus (degrees) was as follows: -0.086 + 0.245 × preoperative esotropia angle (degrees) - 2.025 × amount of IRM nasal transposition (muscle width) (r = 0.542; adjusted $r^2$ = 0.269; P < 0.001) (Fig 2). When the preoperative esotropia angle was large, the postoperative

**Table 1. Patient data and measurement results.**

| | Total | Group A (reduced esotropia) | Group B (unchanged esotropia) | Group C (increased esotropia) | P value |
|---|---|---|---|---|---|
| Number of patients | 62 | 24 (38.7%) | 11 (17.7%) | 27 (43.5%) | |
| Sex (M/F) | 24/38 | 5/19 | 4/7 | 15/12 | 0.039 |
| Affected side (R/L) | 30/32 | 10/14 | 6/5 | 14/13 | 0.694 |
| Age (years) | 60.7 ± 9.5 | 59.6 ± 9.8 | 58.0 ± 7.7 | 62.8 ± 9.6 | 0.281 |
| Smoking status | | | | | |
| 0 (non-smoker) | 46 | 17 | 8 | 21 | 0.752 |
| 1 (<10 cigarettes/day) | 2 | 1 | 0 | 1 | |
| 2 (11–20 cigarettes/day) | 6 | 4 | 1 | 1 | |
| 3 (>20 cigarettes/day) | 8 | 2 | 2 | 4 | |
| History of corticoid therapy/orbital radiotherapy (Y/N) | 27/35 | 10/14 | 8/3 | 9/18 | 0.082 |
| Amount of IRM recession (mm) | 4.8 ± 2.1 | 5.4 ± 2.1 | 4.1 ± 2.0 | 4.6 ± 2.1 | 0.177 |
| Amount of IRM nasal transposition (muscle width) | 0.5 ± 0.4 | 0.4 ± 0.3 | 0.5 ± 0.3 | 0.6 ± 0.4 | 0.103 |
| Preoperative ocular deviation angle (degrees) | | | | | |
| Hypotropia | 12.4 ± 5.7 | 14.1 ± 5.8 | 10.2 ± 5.4 | 11.7 ± 5.4 | 0.124 |
| Esotropia | 3.0 ± 6.1 | 5.3 ± 4.8 | 3.7 ± 5.6 | 0.6 ± 6.4 | 0.019 |
| Excyclotropia | 5.4 ± 4.4 | 4.5 ± 4.4 | 5.0 ± 4.8 | 6.3 ± 4.0 | 0.401 |
| Magnitude of pattern strabismus | -0.8 ± 2.4 | 0 ± 2.1 | -0.7 ± 1.3 | -1.5 ± 2.8 | 0.081 |
| Cross-sectional area (mm$^2$) | | | | | |
| Ipsilateral IRM | 81.2 ± 23.5 | 82.0 ± 20.2 | 80.5 ± 24.3 | 80.8 ± 26.2 | 0.978 |
| Ipsilateral SRM | 27.3 ± 11.1 | 27.0 ± 12.3 | 27.0 ± 11.6 | 27.8 ± 9.9 | 0.966 |
| Ipsilateral MRM | 40.2 ± 9.6 | 41.2 ± 10.4 | 40.2 ± 11.1 | 39.2 ± 8.2 | 0.767 |
| Ipsilateral LRM | 124.1 ± 32.7 | 125.5 ± 30.1 | 112.1 ± 36.9 | 127.8 ± 32.5 | 0.401 |
| Ipsilateral SOM | 21.3 ± 6.9 | 19.1 ± 5.8 | 20.2 ± 5.0 | 23.8 ± 7.7 | 0.051 |
| Ipsilateral IOM | 29.8 ± 10.1 | 28.5 ± 10.3 | 31.6 ± 11.8 | 30.4 ± 9.2 | 0.626 |
| Contralateral IRM | 48.5 ± 17.1 | 48.7 ± 15.2 | 42.6 ± 14.9 | 50.8 ± 19.2 | 0.410 |
| Postoperative ocular deviation angle (degrees) | | | | | |
| Hypotropia | 1.7 ± 3.4 | 2.2 ± 4.5 | 0.8 ± 2.5 | 1.6 ± 2.4 | 0.513 |
| Esotropia | 3.3 ± 5.3 | 2.6 ± 4.3 | 3.7 ± 5.6 | 3.8 ± 5.9 | 0.707 |
| Excyclotropia | -0.3 ± 1.4 | -0.3 ± 1.3 | 0 ± 1.1 | -0.4 ± 1.5 | 0.737 |
| Magnitude of pattern strabismus | -0.6 ± 2.0 | 0.2 ± 1.7 | -0.4 ± 1.1 | -1.2 ± 2.2 | 0.035 |
| Postoperative changes in magnitude of horizontal strabismus (degrees) | -0.4 ± 3.2 | 2.7 ± 1.8 | 0 | -3.2 ± 2.0 | |

M, male; F, female; R, right; L, left; Y, yes; N, no; IRM, inferior rectus muscle; SRM, superior rectus muscle; MRM, medial rectus muscle; LRM, lateral rectus muscle; SOM, superior oblique muscle; IOM, inferior oblique muscle.

Magnitude of pattern strabismus was calculated as subtraction of the horizontal ocular deviation angle measured in 15° downward gaze from that measured in 15° upward gaze.

esotropia tended to decrease; in contrast, when the amount of IRM nasal transposition was large, the postoperative esotropia tended to increase. There was no multi-collinearity between the preoperative esotropia and the amount of IRM nasal transposition (variance inflation factor, 1.003).

There was little correlation between the amount of IRM recession and that of IRM nasal transposition (r = 0.147, P = 0.255).

All the patients were in the burnout phase of TED and no patient showed worsening of CAS more than 3 points postoperatively. One patient in Group A underwent left medial rectus muscle recession for residual esotropia. None of the patients in Group C were treated for

**Table 2. Statistical results of univariate and following multivariate linear regression analysis with stepwise variable selection.**

| Predictive factors | | Univariate | | Multivariate stepwise | |
|---|---|---|---|---|---|
| | | *P* value | Crude coefficient (95% CI) | *P* value | Crude coefficient (95% CI) |
| Age | | 0.466 | -0.033 (-0.119 to 0.053) | - | - |
| Smoking status | | 0.787 | 0.100 (-0.637 to 0.838) | - | - |
| History of corticoid therapy/orbital radiotherapy | | 0.655 | 0.366 (-1.265 to 1.997) | - | - |
| Amount of IRM recession | | 0.499 | 0.132 (-0.256 to 0.519) | - | - |
| Amount of IRM nasal transposition | | 0.041 | -2.230 (-4.558 to -0.101) | 0.049 | -2.025 (-4.043 to 0.008) |
| Angle of preoperative esotropia | | < 0.001 | 0.254 (0.137 to 0.372) | < 0.001 | 0.245 (0.129 to 0.361) |
| Cross-sectional area | | | | | |
| | Ipsilateral IRM | 0.962 | -0.001 (-0.036 to 0.034) | - | - |
| | Ipsilateral SRM | 0.490 | -0.025 (-0.099 to 0.048) | - | - |
| | Ipsilateral MRM | 0.928 | 0.004 (-0.081 to 0.089) | - | - |
| | Ipsilateral LRM | 0.197 | -0.016 (-0.041 to 0.009) | - | - |
| | Ipsilateral SOM | 0.012 | -0.147 (-0.259 to -0.034) | - | - |
| | Ipsilateral IOM | 0.283 | -0.044 (-0.126 to 0.037) | - | - |
| | Contralateral IRM | 0.523 | -0.015 (-0.063 to 0.032) | - | - |

CI, confidence interval; IRM, inferior rectus muscle; SRM, superior rectus muscle; MRM, medial rectus muscle; LRM, lateral rectus muscle; SOM, superior oblique muscle; IOM, inferior oblique muscle.

increased postoperative esotropia with prisms or additional surgery. Based on our clinical evidence, none of the patients developed a slipped IRM postoperatively.

## Discussion

The present study demonstrated that the changes in horizontal strabismus after IRM recession ± IRM nasal transposition were not uniform in patients with TED. Several possible influential factors for the postoperative changes were identified.

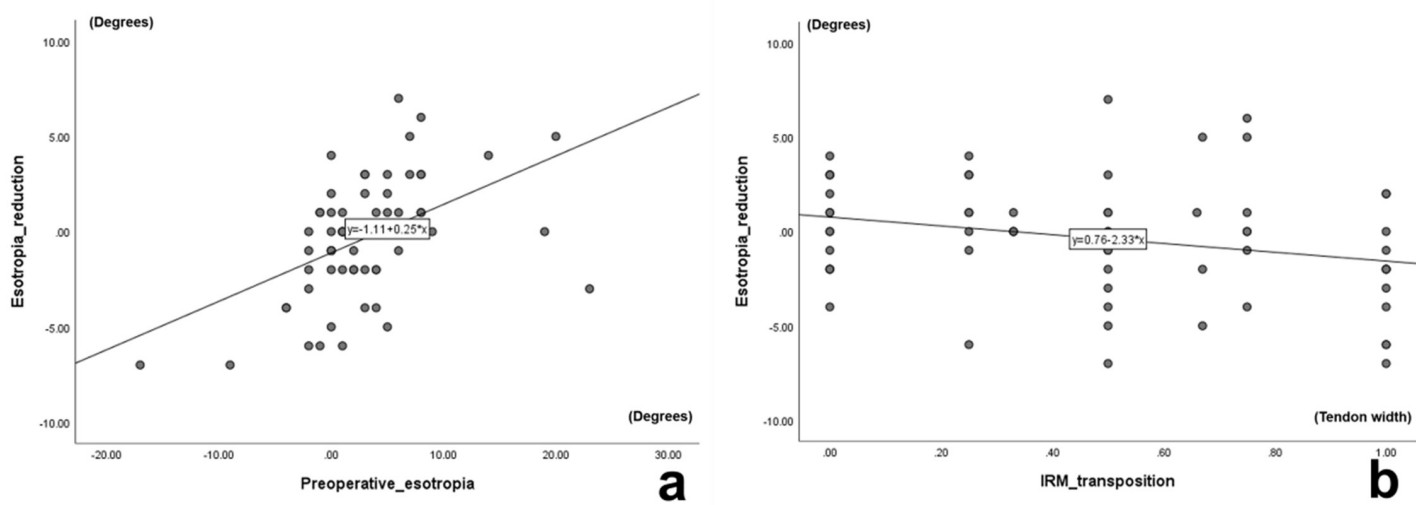

**Fig 2. Scatter plots. a.** The relationship between preoperative esotropia (x-axis) and esotropia reduction (y-axis). **b.** The relationship between amount of inferior rectus muscle (IRM) transposition (x-axis) and esotropia reduction (y-axis).

Multiple regression analyses identified that the postoperative reduction in the magnitude of esotropia had a significant positive correlation with the angle of preoperative esotropia ($P < 0.001$). Also, the angle of preoperative esotropia was significantly larger in patients of Group A ($5.3 \pm 4.8$ degrees) than those of Group C ($0.6 \pm 6.4$ degrees; $P = 0.015$). Although a considerable number of patients in TED show combined hypotropia and esotropia because of the frequent involvement of the IRM and MRM, these results suggest that patients with a larger angle of preoperative esotropia are expected to have a better improvement in esotropia postoperatively.

On the contrary, the amount of IRM nasal transposition was found to be negatively correlated with the postoperative reduction in esotropia. This may result from the greater adduction force of a nasally transposed IRM compared to the originally positioned IRM, because the nasally transposed IRM approaches the MRM such that the duction direction of the IRM becomes similar to that of the MRM.

Based on the fact that the IRM is a secondary adductor, in about $1/3^{rd}$ of the patients, esotropia was found to be reduced postoperatively, which was expected. However, 43.5% of the patients showed opposite results in which the esotropia was increased postoperatively. One of the possible reasons for this discrepancy may be a thicker SOM in patients of Group C. Similarly, a negative correlation was also seen between the postoperative changes in the magnitude of esotropia and cross-sectional area of the SOM. In TED, the SOM is found to be occasionally enlarged [18]. As the SOM is not the antagonist of IRM, the function of SOM may little change after IRM recession and its nasal transposition. Therefore, the thicker SOM may mainly affect only the preoperative status of the involved eye. As the SOM is an abductor, we can infer that a thicker SOM in patients of Group C may be associated with a smaller angle of preoperative esotropia. From another point of view, the incycloduction force of the thicker SOM may also counteract the preoperative excyclotropia. This might result in less amount of IRM nasal transposition, although the preoperative excyclotropia and the amount of IRM nasal transposition were not significantly different among the groups.

The magnitude of pattern strabismus was not different between the pre- and postoperative ones in total and in each group, respectively. A previous study demonstrated that the bilateral IRM recession increases A-pattern esotropia in TED [13]. However, the results in this study indicate that IRM recession ± IRM nasal transposition only on one side did not increase the risk of pattern strabismus.

Although the ratio of patients who had undergone corticoid therapy and/or orbital radiotherapy before surgery tended to be larger in Group B, this factor was not a significant predictor of postoperative changes in horizontal strabismus. Our previous studies also showed that a history of orbital radiotherapy did not significantly affect the correlation between nasal IRM transposition and excyclotropia correction, as well as the dose-effect relationship between IRM recession and hypotropia correction [12,19]. The combined results of our present and previous reports indicate that a history of such anti-inflammatory treatment does not affect the surgical results of IRM recession ± IRM nasal transposition.

Smoking status was not a significant predictive factor of the changes in postoperative horizontal strabismus. Smokers usually exhibit a higher degree of TED severity [7,8,11,23]. On the contrary, smokers exhibited a higher degree of adipose change with the same IRM thickness as compared with non-smokers [12]. This causes less fibrous changes in the IRM of smokers, resulting in less contraction force of the eye towards the medial direction. Therefore, these features may offset each other, resulting in no significance in this study.

The number of male patients was significantly larger in Group C. As male patients commonly exhibit more severe manifestations of TED [26], these may be associated with the presence of a thicker SOM in Group C, as found in this study.

The right-to-left ratio and the patient age at the time of surgery were not significantly different between the groups. These results tend to suggest that the laterality and the patient age do not influence the changes in horizontal strabismus after surgery.

A previous study by Kim and Kim found that the changes in horizontal strabismus after SRM recession were nonuniform and that among 39 patients, 14 showed exodeviation, 18 esodeviation, and 7 no change, although the changes did not reach statistical significance [16]. Furthermore, they did not investigate the predictive factors and included non-TED patients as well. However, the results in that study were somewhat similar to those in the present study.

None of the patients in Group C were treated for increased esotropia with prisms or additional surgery. In Group C, although the esotropia increased after surgery, the preoperative horizontal deviation angle was very small (0.6 ± 6.4˚), such that it did not result in any symptomatic horizontal diplopia after surgery.

We used a synoptophore to measure the ocular deviation angle as the angle of cyclotropia could be measured with the head fixed position. Also, the association between the degree of correction of hypotropia and excyclotropia had been previously clarified with this measuring method [12,19]. Although the horizontal angle of ocular deviation might be disturbed by the instrumental accommodation during the synoptophore test, our patients were old (mean, 60.7 years) enough to have minimal instrumental accommodation. Besides, fixation of the surgical eye during the synoptophore test resulted in a measurement of a larger deviation angle, which was in accordance with the Hering's law [12]. However, as hypotropia and excyclotropia were well corrected after surgery, this overshoot might have led to better preoperative estimation.

The main methodological limitation of this study was its retrospective design. Besides, we included patients with TED who did not have a history of orbital decompression surgery which causes more restriction of extraocular muscle motility [11,27]. Therefore, the present regression model cannot be applied to those patients with TED who have a history of orbital decompression surgery. Future studies are necessary to determine the regression models for such patients. Furthermore, measurement of the volume of each extraocular muscle might have provided a more precise statistical analysis in this study. Although we did not measure the abduction deficit, alternatively the cross-sectional area of the MRM was measured, which reflected the abduction deficit. Also, we did not perform a forced duction test at the start and the end of the surgery. These data might have provided more information on the changes in the magnitude of esotropia. Moreover, the deviation angles were measured by a single orthoptist. Measurements by multiple examiners and use of mean measurement values may provide more accurate outcomes.

## Conclusions

Although the changes in horizontal strabismus after IRM recession ± IRM nasal transposition were nonuniform in patients with TED, patients with a larger angle of preoperative esotropia are expected to have a better improvement in esotropia after surgical correction of hypotropia. In addition, a less amount of IRM nasal transposition may be associated with a more postoperative reduction in the angle of esotropia. We believe the results of our study will be helpful to ophthalmologists for formulating an effective preoperative surgical plan.

## Supporting information

**S1 Fig. Intraoperative photos. a.** After securing the inferior rectus muscle at its insertion with a muscle hook, the muscle tendon is secured using locking 8–0 polyglactin sutures at two points, 1 mm posterior to the globe insertion. **b.** The inferior rectus muscle is recessed and

fixed onto the sclera.
(TIF)

**S1 File. The raw data.**
(XLSX)

## Acknowledgments

The authors wish to be thankful to Dr. Yoshiyuki Kitaguchi at Department of Ophthalmology, Osaka University, and Ms. Naomi Umezawa, an orthoptist at Aichi Medical University Hospital, for her help to collect and interpret the data.

## Author Contributions

**Conceptualization:** Yasuhiro Takahashi.

**Data curation:** Yasuhiro Takahashi.

**Formal analysis:** Yasuhiro Takahashi.

**Investigation:** Yasuhiro Takahashi.

**Methodology:** Yasuhiro Takahashi.

**Project administration:** Yasuhiro Takahashi.

**Supervision:** Hirohiko Kakizaki.

**Writing – original draft:** Aric Vaidya.

**Writing – review & editing:** Hirohiko Kakizaki, Yasuhiro Takahashi.

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
