## [Decision Letter · Decision Letter 0]

24 Aug 2020

PONE-D-20-20308

Changes in Horizontal Strabismus after Inferior Rectus Muscle Recession with or without Nasal Transposition in Thyroid Eye Disease: A Retrospective, Observational Study

PLOS ONE

Dear Dr. Takahashi,

Thank you for submitting your manuscript to PLOS ONE. After careful consideration, we feel that it has merit but does not fully meet PLOS ONE’s publication criteria as it currently stands. Therefore, we invite you to submit a revised version of the manuscript that addresses the points raised during the review process.

We look forward to receiving your revised manuscript.

Kind regards,

Ahmed Awadein, MD, Ph.D, FRCS

Academic Editor

PLOS ONE

Journal Requirements:

2. We ask that you please consider moving Figure 2 to Supporting Information due to is graphic nature.

Reviewers' comments:

Reviewer's Responses to Questions

**Comments to the Author**

1. Is the manuscript technically sound, and do the data support the conclusions?

Reviewer #1: Partly

Reviewer #2: Yes

2. Has the statistical analysis been performed appropriately and rigorously? 

Reviewer #1: I Don't Know

Reviewer #2: Yes

3. Have the authors made all data underlying the findings in their manuscript fully available?

Reviewer #1: Yes

Reviewer #2: Yes

4. Is the manuscript presented in an intelligible fashion and written in standard English?

Reviewer #1: Yes

Reviewer #2: Yes

5. Review Comments to the Author

Reviewer #1: In the preoperative data the lack of forced diction test especially on the MRM which may largely contribute to the esotropia is a defect in the study, the authors admitted that.

In group C where the esotropia increases postoperatively the preoperative angles of esotropia were very small (lines 358-360) and the mean of pre and post. operative angle does not show how much is the increase of the angle postoperatively. There might be a fallacy in measurement of small angles of deviation. In such cases the angles should be measured by more than one examiner and an average taken to be sure of the results and conclusions.

The explanation of the role of SOM contribution to the increase of angle of esotropia in group C ( lines 313-317) is not justified. Decreasing the amount of IRM recession and transposition according to the conclusion of the study should decrease the angle of Esotropia not increase it as in group C.

The relation between the amount of IRM recession and its nasal transposition should be studied together with its effect on esotropia. IRM recession decreases esotropia ( weakening an adductor) while its nasal transposition may increase esotropia both together may nullify each other or one effect may be more than the other. The resultant change in esotropia May be explained in that way.

Reviewer #2: Methods: The details of the radiotherapy and MRI are not relevant to the study and can be trimmed

172- 8/0 suture is too thin for eye muscle surgery. Can you comment on that?

198-200 How were measurements taken in upgaze and downgaze?

202-205 and throughout the manuscript replace subtraction values by the magnitude of pattern strabismus

358 preoperative or postoperative?

Can you add a line to the scatter plots and include the regression equation in the figure?

6. PLOS authors have the option to publish the peer review history of their article (what does this mean?). If published, this will include your full peer review and any attached files.

Reviewer #1: **Yes: **Lobna khazbak, MD

Reviewer #2: **Yes: **Ahmed Awadein

---

## [Author Response · Author response to Decision Letter 0]

5 Sep 2020

Point-by-Point Response to Reviewers’ Comments

We wish to thank the reviewers for giving their precious time to help us improve our manuscript. We value all the comments that were given to us and hope that the changes we made will make the manuscript more acceptable to the reviewers.

Journal Requirements:

Comment 1: Please ensure that your manuscript meets PLOS ONE's style requirements, including those for file naming. The PLOS ONE style templates can be found at

Reply: We revised the paper according to PLOS ONE’s style requirements.

Comment 2: We ask that you please consider moving Figure 2 to Supporting Information due to its graphic nature.

Reply: We moved the original Figure 2 to S1 Figure and moved up the original Figure 3 to Figure 2.

Comment 3: Please include captions for your Supporting Information files at the end of your manuscript, and update any in-text citations to match accordingly. Please see our Supporting Information guidelines for more information: http://journals.plos.org/plosone/s/supporting-information

Reply: We included the captions for the supporting information files at the end of the manuscript.

 

Response to Comments from Reviewer #1:

Comment #1: In the preoperative data the lack of forced diction test especially on the MRM which may largely contribute to the esotropia is a defect in the study, the authors admitted that.

Reply: Thank you for your suggestion. As pointed out by the reviewer, we did not perform a forced duction test at the start and the end of the surgery, which is a limitation of our study. We stated this limitation in page 19, lines 366-368.

Comment #2: In group C where the esotropia increases postoperatively the preoperative angles of esotropia were very small (lines 358-360) and the mean of pre and post operative angle does not show how much is the increase of the angle postoperatively.

Reply: Thank you for your suggestion. We presented the changes in the horizontal angle in Table 1.

Comment #3: There might be a fallacy in measurement of small angles of deviation. In such cases the angles should be measured by more than one examiner and an average taken to be sure of the results and conclusions.

Reply: Thank you for pointing this out. As indicated by the reviewer, the deviation angles were measured by a single orthoptist in this study. Measurements by multiple examiners and use of mean measurement values may provide more accurate outcomes. We added this limitation in page 19, line 368 to page 20, line 370.

Comment #4: The explanation of the role of SOM contribution to the increase of angle of esotropia in group C (lines 313-317) is not justified. Decreasing the amount of IRM recession and transposition according to the conclusion of the study should decrease the angle of Esotropia not increase it as in group C.

Reply: Thank you for your indication. 

We had discussed about the contribution of a thicker SOM to the magnitude of preoperative esotropia and excyclotropia because the SOM is not the antagonist of the IRM, and the function of the SOM little change after IRM recession and its nasal transposition. With regard to the above points, we believe that our discussion is reasonable. We added the following sentences in page 16, lines 297-299:

“As the SOM is not the antagonist of IRM, the function of SOM may not change after IRM recession and its nasal transposition. Therefore, the thicker SOM may mainly affect only the preoperative status of the involved eye.” 

As commented by the reviewer, firstly, we had expected that IRM recession ± IRM nasal transposition always reduces esotropia. Surprisingly, instead of postoperative reduction in esotropia, we happened to experience increased esotropia after IRM recession ± IRM nasal transposition, as shown in Group C. We, therefore, studied the tendency of postoperative changes in esotropia after performing such surgery. We added the following sentence on page 3, lines 42-44: “Moreover, in some cases, contrary to the expected postoperative reduction in esotropia, we happened to experience increased esotropia following surgery.”

Comment #5: The relation between the amount of IRM recession and its nasal transposition should be studied together with its effect on esotropia. IRM recession decreases esotropia (weakening an adductor) while its nasal transposition may increase esotropia both together may nullify each other or one effect may be more than the other. The resultant change in esotropia May be explained in that way.

Reply: Thank you very much for your suggestion. We analyzed the relationship between the amount of IRM recession and that of IRM nasal transposition using Pearson’s correlation coefficient, but there was little correlation between them (r = 0.147, P = 0.255). We added these contents in page 9, lines 192-195, and page 15, lines 261-262. 

 

Response to Comments from Reviewer #2:

Comment #1: Methods: The details of the radiotherapy and MRI are not relevant to the study and can be trimmed.

Reply: Thank you for your suggestion. We shortened the corresponding parts in pages 5-6. 

Comment #2: 172- 8/0 suture is too thin for eye muscle surgery. Can you comment on that?

Reply: We used 8/0 suture because of our preference only. As pointed out by the reviewer, we agree that this suture is thin. We, therefore, fixed the inferior rectus muscle tendon at 4-6 points to prevent the slippage of the muscle. Actually, based on our clinical evidence, none of the patients developed a slipped IRM postoperatively. We stated these contents in page 8, lines 161-162, and page 15, lines 267-268.

Comment #3: 198-200 How were measurements taken in upgaze and downgaze?

Reply: We measured the deviation angles in 15° upward gaze and 15° downward gaze using a synoptophore. One of the two arms of the synoptophore was fixed at 15° upward gaze or 15° downward gaze, and then, the deviation angle of the non-fixating eye was measured. We stated these contents in page 6, 2nd paragraph, lines 113-118.

Comment #4: 202-205 and throughout the manuscript replace subtraction values by the magnitude of pattern strabismus

Reply: Thank you for your suggestion. We modified the manuscript according to the reviewer’s comment.

Comment #5: 358 preoperative or postoperative?

Reply: Thank you for pointing this out. The angle shown in the corresponding sentence was measured preoperatively. 

Comment #6: Can you add a line to the scatter plots and include the regression equation in the figure?

Reply: Thank you for your suggestion. We added the regression lines in the new Figure 2 (the original Figure 3).

---

## [Editor Report · Decision Letter 1]

18 Sep 2020

Changes in horizontal strabismus after inferior rectus muscle recession with or without nasal transposition in thyroid eye disease: a retrospective, observational study

PONE-D-20-20308R1

Dear Dr. Takahashi,

We’re pleased to inform you that your manuscript has been judged scientifically suitable for publication and will be formally accepted for publication once it meets all outstanding technical requirements.

Kind regards,

Ahmed Awadein, MD, Ph.D, FRCS

Academic Editor

PLOS ONE
---

## [Editor Report · Acceptance letter]

22 Sep 2020

PONE-D-20-20308R1 

Changes in horizontal strabismus after inferior rectus muscle recession with or without nasal transposition in thyroid eye disease: a retrospective, observational study 

Dear Dr. Takahashi:

I'm pleased to inform you that your manuscript has been deemed suitable for publication in PLOS ONE. Congratulations! Your manuscript is now with our production department. 

Kind regards, 

on behalf of

Dr. Ahmed Awadein 

Academic Editor

PLOS ONE